# Engineering a Phosphoketolase Pathway to Supplement Cytosolic Acetyl-CoA in *Aspergillus niger* Enables a Significant Increase in Citric Acid Production

**DOI:** 10.3390/jof9050504

**Published:** 2023-04-23

**Authors:** Jiao Liu, Shanshan Zhang, Wenhao Li, Guanyi Wang, Zhoujie Xie, Wei Cao, Weixia Gao, Hao Liu

**Affiliations:** 1MOE Key Laboratory of Industrial Fermentation Microbiology, College of Biotechnology, Tianjin University of Science & Technology, Tianjin 300457, China; liuj@tust.edu.cn (J.L.);; 2Tianjin Engineering Research Center of Microbial Metabolism and Fermentation Process Control, Tianjin University of Science & Technology, Tianjin 300457, China; 3National Technology Innovation Center of Synthetic Biology, Tianjin 300308, China

**Keywords:** *Aspergillus niger*, acetyl-CoA, citrate, phosphoketolase

## Abstract

Citric acid is widely used in the food, chemical and pharmaceutical industries. *Aspergillus niger* is the workhorse used for citric acid production in industry. A canonical citrate biosynthesis that occurred in mitochondria was well established; however, some research suggested that the cytosolic citrate biosynthesis pathway may play a role in this chemical production. Here, the roles of cytosolic phosphoketolase (PK), acetate kinase (ACK) and acetyl-CoA synthetase (ACS) in citrate biosynthesis were investigated by gene deletion and complementation in *A. niger*. The results indicated that PK, ACK and ACS were important for cytosolic acetyl-CoA accumulation and had significant effects on citric acid biosynthesis. Subsequently, the functions of variant PKs and phosphotransacetylase (PTA) were evaluated, and their efficiencies were determined. Finally, an efficient PK-PTA pathway was reconstructed in *A. niger* S469 with *Ca*-PK from *Clostridium acetobutylicum* and *Ts*-PTA from *Thermoanaerobacterium saccharolyticum*. The resultant strain showed an increase of 96.4% and 88% in the citrate titer and yield, respectively, compared with the parent strain in the bioreactor fermentation. These findings indicate that the cytosolic citrate biosynthesis pathway is important for citric acid biosynthesis, and increasing the cytosolic acetyl-CoA level can significantly enhance citric acid production.

## 1. Introduction

Citric acid, a significant organic acid, has widespread use in the food, chemical and pharmaceutical industries [1,2]. In the past century, *Aspergillus niger* has become the main workhorse for industrial citric acid production [1,2]. The canonical view is that the citric acid biosynthesis in *A. niger* primarily occurs via the mitochondrial pathway, as depicted in Figure 1A [3]. Briefly, citric acid is synthesized by the citrate synthase (CitA) in mitochondria using acetyl-CoA and oxaloacetate as precursors. Acetyl-CoA and oxaloacetate are produced from the cytoplasm-introduced pyruvate and malate, respectively. The exchange of citrate from the mitochondria to the cytoplasm and malate from the cytoplasm to the mitochondria is facilitated by the mitochondrial citrate/malate antiporter.

Industrial strains of *A. niger* used for citric acid production are usually acquired through conventional mutagenesis techniques, such as chemical and UV-based random mutation [1,2,4]. When applying these conventional techniques, a time-consuming high-throughput screening step is often followed, and the resulting strains frequently suffer from genetic instability [1,2,4]. To address the drawbacks of conventional mutagenesis, various rational metabolic engineering approaches have been devised for improving citric acid production in *A. niger*, based on the knowledge of the mitochondrial citric acid biosynthesis pathway [2,4,5]. Enhancing citric acid efflux and reducing ATP biosynthesis could significantly improve citric acid production in *A. niger* through the overexpression of the citric acid transporters CexA and the alternative oxidase (AOX) [6,7,8]. However, the strategies, involving overexpressing the *citA* and enhancing the availability of the precursor oxaloacetate and acetyl-CoA, have not been successful in significantly increasing citric acid production [9,10,11,12]. *A. niger*’s synthesis of citric acid is regulated by a complex mechanism. In some cases, unexpected results have been reported [4]. For instance, increasing the consumption of cytosolic citrate through enhancing the ATP-citrate lyase (ACL) pathway has been found to result in a boost in citric acid production in *A. niger* [4,12]. These findings led to the hypothesis that an unknown cytosolic citrate biosynthesis pathway may exist in *A. niger* [4]. The acetyl-CoA present in the cytosol may directly participate in the cytosolic biosynthesis of citric acid in *A. niger*. In our previous study, the overexpression of the gene *acl* encoding ACL augmented cytosolic acetyl-CoA, which may be the primary driver of enhanced citric acid biosynthesis in the modified strain. The subsequent deletion of *acl* resulted in decreased cytosolic acetyl-CoA and citric acid, supporting the above hypothesis [12]. This hypothesis was supported by further evidence, such as genomic research revealing the presence of genes encoding cytosolic citrate synthase (e.g., CitB) in the genome of *A. niger* and studies that demonstrated the functionality of CitB in various fungi [13,14]. For example, in the study conducted by Hossain et al., it was demonstrated that overexpressing the gene *citB* in an engineered *A. niger* strain led to an increase in itaconic acid production [13]. Citric acid was the direct precursor for itaconic acid biosynthesis. In a separate study by Vesth et al., introducing the *citB* gene from *A. niger* into *Aspergillus nidulans*, which does not naturally produce citric acid, resulted in the accumulation and production of citric acid in the transformed strain [14].

**Figure 1 jof-09-00504-f001:**
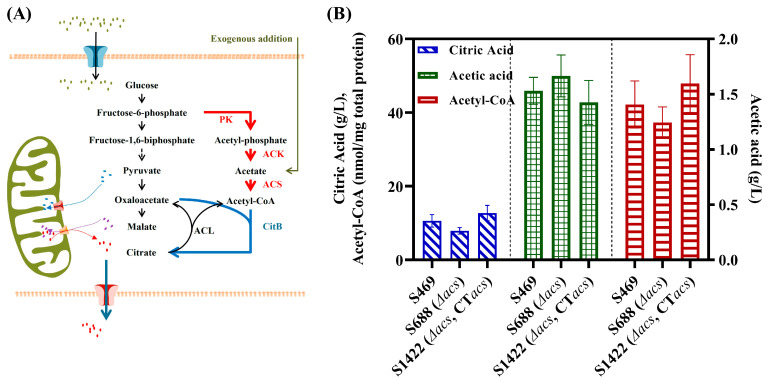
Roles of ACS in the synthesis of acetyl-CoA and citric acid in *A. niger*. (**A**) The metabolic pathway of citric acid in *A. niger*; the black arrows represent the canonical pathway of citric acid metabolism, the blue arrow means the putative cytosolic citrate synthesis pathway, the red arrows indicate the putative complementary pathway of cytosolic acetyl-CoA and the green arrow indicates that acetic acid addition can supplement the cytosolic acetyl-CoA in the *acl*-deletion strain [7]. (**B**) The concentrations of extracellular citric and acetic acid and intracellular acetyl-CoA of S469, S688 and S1422 during a 3-day citrate fermentation in a shake flask at 28 °C and 200 rpm. CT: complementation of ACK driven by the *gpdA* promoter at the *amyA* locus of S688; ACK: acetate kinase; ACS: acetyl-CoA synthetase; CitA: mitochondrial citrate synthase; CitB: cytosolic citrate synthase; PK: phosphoketolase.

Levente Karaffa and Christian P Kubicek had speculated that acetic acid can be converted by acetyl-CoA synthase (ACS) to acetyl-CoA, utilized for the cytosolic biosynthesis of citric acid in *A. niger* [3]. The result of our previous research supported this speculation. The deletion of the *acl* gene in *A. niger* resulted in a significant growth impairment, as well as a decline in the levels of both acetyl-CoA and citric acid. However, these effects could be mitigated by adding acetate to the growth medium [12]. Furthermore, a cytosolic phosphoketolase (PK, NCBI-Gene ID: 3512051), converting fructose-6-phosphate to acetyl-phosphate, was found in the filamentous fungus *A. nidulans* [15]. Acetyl-phosphate can then be converted to acetic acid by an enzyme called acetate kinase (ACK). A metabolic network analysis using [1-^13^C] glucose revealed that approximately 29.2% of acetyl-CoA in *A. nidulans* was produced by the PK pathway [15]. The genome of *A. niger* contains close homologs of the PK found in *A. nidulans*, indicating the presence of a similar cytosolic acetyl-CoA biosynthesis pathway in *A. niger* as well.

In this study, we characterized the PK pathway in *A. niger* through genetic methods. Our findings show that the PK pathway was an important source of cytosolic acetyl-CoA, which directly affected citric acid production through the cytosolic citric acid biosynthesis pathway. Moreover, by introducing specific enzymes from other microorganisms, we were able to establish a highly efficient cytosolic PK pathway, resulting in a significant increase in citric acid production. These results offer new possibilities for improving *A. niger* strains for citric acid production.

## 2. Materials and Methods

### 2.1. Strains and Culture Conditions

The parent strain *A. niger* S469 was created by overexpressing the Cyclization Recombination Enzyme (Cre) using the inducible promoter *Tet-on* in *A. niger* ATCC 1015 [16]. Spore preparation, transformant screening and gene knock-out phenotype screening were carried out using Potato dextrose agar (PDA), complete medium (CM) and minimal medium (MM), respectively, as previously described [17]. All *A. niger* strains were grown at 28 °C. *Escherichia coli* JM109 and *Agrobacterium tumefaciens* AGL-1, used for the plasmid construction and genetic transformation of *A. niger*, respectively, were grown in LB medium at 37 °C [16]. For the strain and plasmid construction, 250 μg/mL hygromycin B or 100 μg/mL kanamycin was added to the medium, when required. Citric acid production was performed in a shake flask using a citrate fermentation medium consisting of 10% sucrose, 0.25% NH_4_NO_3_, 0.1% MgSO_4_·7H_2_O, 0.1% KH_2_PO_4_ and 0.05% yeast extract (pH adjusted to 2.5) [18]. The *A. niger* strains used in this study are listed in Table 1.

### 2.2. RNA Isolation and RT-PCR Analysis of acs, ack, pkA and pkB

The mycelia for total RNA isolation were harvested from the citrate fermentation medium during shake flask cultivation. Total RNA was then isolated using the E.A.N.A.TM Fungal RNA Kit (from Omega Bio-tek, Inc., Norcross, GA, USA), as per the manufacturer’s protocol. Complementary DNA (cDNA) was synthesized using the PrimeScript™ II 1st-Strand cDNA Synthesis Kit (from TaKaRa Biomedical Technology Co., Ltd., Beijing, China). The transcription levels of *acs*, *ack*, *pkA* and *pkB* in *A. niger* were determined by performing PCR, using the cDNA as a template. The primer sequences can be found in Appendix A.

### 2.3. Plasmid Construction

In this study, the plasmids utilized can be found in Table 1, and the primer information is provided in Appendix A. These plasmids were generated using the ClonExpress II One-Step Cloning Kit (C112-01/02) from Nanjing Vazyme Biotech Co., Ltd., (Nanjing, China)

Gene disruption plasmids: Using pLH594 as the parent vector, the *acs* (locus_tag: ASPNIDRAFT_214348), *ack* (locus_tag: ASPNIDRAFT_206885), *pkA* (locus_tag: ASPNIDRAFT_197387) and *pkB* (locus_tag: ASPNIDRAFT_54814) disruption plasmids (pLH525, pLH670, pLH1549 and pLH1551, respectively) were built using the same method as reported in a prior study [17]. For instance, the primer pairs acsU-F/R and acsD-F/R were used to amplify the dual flanked regions of the *acs* coding region, using the genomic DNA from *A. niger* S469 as the template. The two fragments were then ligated sequentially into the flanks of the hygromycin resistance cassette (*loxP-hph-loxP*) in pLH594, creating the *acs*-deletion plasmid pLH525.

Plasmids utilized for gene complementation or heterologous expression at the amyA locus: The transcription of the heterologous genes (*Bl-pk* GenBank AAN24771.1, *Ba-pk* NCBI-Gene ID 56674845, *Ca-pk* GenBank KHD36088.1, *Ts-pta* GenBank ADU15860.1 and *Bs-pta* NCBI-Gene ID 936581) and the native genes (*acs*, *ack*, *pkA* and *pkB*) was driven by the *gpdA* promoter in these plasmids. The coding sequences of native genes were amplified from cDNA of S469, whereas heterologous genes were artificially synthesized after codon optimization. The plasmids used for the gene complementation or heterologous expression of these genes was created with the same method. For example, the *acs* coding gene was acquired using the primer pair acs-F/R, ligated into pLH454 to create pLH821, and then the *acs* expression cassette with the *gpdA* promoter and *trpC* terminator was amplified from pLH821 using the primer pair P924-454-L/R. To create pLH822, the resulting fragment was joined to pLH924. The pLH822 vector was used for the gene complementation of *acs*.

Plasmids used for the co-expression of *Ca-pk* and *Ts-pta*: The plasmid pLH1080 was used to create the *Ca-pk* expression cassette, featuring the *gdhA* promoter and the *trpC* terminator. This plasmid was derived from pLH454 by replacing the *gpdA* promoter with the *gdhA* promoter, and its sequence can be found in the Appendix A. The *Ca-pk* coding gene was amplified using the primer pair PgdhACapk-L/R and inserted into pLH1080 to form pLH1804. The *Ca-pk* expression cassette was then amplified using the primer pair P1804-L/R and ligated into pLH1589 to produce pLH1805. The pLH1805 vector was used for the co-expression of *Ca-pk* and *Ts-pta*.

### 2.4. Construction of Strains

The genetic manipulation of *A. niger* was carried out using a previously described *Agrobacterium*-mediated transformation and Cre-*lox*P-based genetic system [16]. The *Agrobacterium*-mediated transformation was utilized to introduce the plasmid into *A. niger*, and the Cre-*lox*P-based genetic system was employed for the recombination of *loxP-hph-loxP* loci, resulting in the *hph*-excision strain.

Gene disruption mutants: The *acs*, *ack*, *pkA* and *pkB*-disruption mutants were created using the homologous recombination-based one-step gene replacement method, as previously disclosed by Cao et al. (2020) [17]. As an illustration, the specific procedure for *acs* disruption was as follows. Plasmid pLH525 was introduced into *A. niger* S469, and transformants were obtained on CM plates supplemented with cefotaxime sodium (100 μg/mL), hygromycin B (250 μg/mL), ampicillin (100 μg/mL) and streptomycin (100 μg/mL) at 28 °C for 5 days. Putative disruption mutants were identified as hygromycin B-resistant and glufosinate ammonium-sensitive transformants, which were then isolated using PDA plates with hygromycin B (250 μg/mL) and MM with glufosinate ammonium (1000 μg/mL). The confirmation of the Δ*acs* mutants was carried out using PCR analysis with four primer pairs (acsKO-1/2, acsKO-3/4, acsKO-1/P641, acsKO-3/P642 and acsKO-1/4) (as shown in Appendix A). The verified mutant was named S687, and then the *hgh* was excised with a Cre-*lox*P-based genetic system to obtain S688. The same method was used to generate additional mutants: S917 (Δ*ack*), S2738 (Δ*pkA*), S2908 (Δ*pkB*) and S3050 (Δ*pkA*Δ*pkB*) (as shown in Appendix A). The S3050 was obtained by excising the *hph* in S3008, which was created by introducing pLH1549 into S2908. The verification of the *hph*-excision strains was carried out using the hph-F/R primer pair.

Strains for gene complementation or heterologous expression: The complementation of the native genes (*acs*, *ack*, *pkA* and *pkB*) and the expression of heterologous genes (*Bl-pk*, *Ba-pk*, *Ca-pk*, *Ts-pta* and *Bs-pta*) were achieved through homologous recombination, using the same strategy as was used for gene disruption. For example, after introducing plasmid pLH822 into the S688 strain, the *acs* expression cassette was integrated into the *amyA* locus, resulting in the S1422 strain. The strains S1724, S2909 and S2914 were generated by introducing the plasmids pLH835, pLH1593 and pLH1594 into S917, which contained the *ack*, *Ts-pta* and *Bs-pta* gene expression cassettes, respectively. By transforming plasmids harboring the *pkA*, *pkB*, *Bl-pk*, *Ba-pk* and *Ca-pk* gene expression cassettes into the S3050 strain, respectively, the following strains were created: S3098, S3104, S3081, S3087 and S3092.

Strain for the co-expression of Ca-pk and *Ts-pta*: The plasmid pLH1805, carrying the co-expression cassette of *Ca-pk* and *Ts-pta*, was introduced into strain S469. Transformants were subsequently screened on PDA with 250 g/mL hygromycin B. The incorporation of the *Ca-pk* and *Ts-pta* co-expression cassette was then verified via PCR analysis using the primer pairs PgdhACapk-L/R and Ts-pta-L/R. Through non-homologous recombination, the *Ca-pk* and *Ts-pta* co-expression cassette could be arbitrarily inserted into various loci of the genome. As a result, a total of 11 verified strains were chosen for citric acid production. *A. niger* S3158, the strain with the highest output, was named.

### 2.5. Citric Acid Shake-Flask Fermentation and Bioreactor Fermentation

Shake-flask fermentation: as previously described, 2 × 10^6^ conidia/mL of *A. niger* mutants were inoculated into 50 mL of citrate fermentation medium in 250 mL Erlenmeyer flasks and incubated at 28 °C and 200 rpm for 3 days [18].

Bioreactor fermentation: seed cultivation was first performed with the same medium and conditions used for shake-flask fermentation. After 24 h of cultivation, the culture at 10% of the volume was inoculated into a 2 L bioreactor containing 1.26 L of citrate fermentation medium. The final volume of the fermentation liquid reaches 70% of the bioreactor capacity. The bioreactor fermentation was performed at a 1.2 vvm aeration rate, 250 r/min and 28 °C for 120 h.

### 2.6. Intracellular Acetyl-CoA Concentration Measurement

The measurement of the intracellular acetyl-CoA level was performed using an Acetyl-CoA Assay Kit (Solarbio, BC0980), and the total protein concentration was evaluated with a BCA Protein Assay Kit (TaKaRa, T9300A) according to the manufacturer’s instructions, as previously described [18].

### 2.7. HPLC Analysis of Citric Acid and Acetic Acid

Citric acid and acetic acid were qualitatively determined by HPLC (Agilent 1260 Infinity II Prime) equipped with an organic acid column (Aminex HPX-87H, 300 mm × 7.8 mm), as previously described [16]. The column temperature was controlled at 65 °C. The mobile phase was 5 mM H_2_SO_4_, with a flow rate at 0.6 mL/min. The UV detection wavelength was set at 210 nm.

### 2.8. Determination of the Cell Dry Weight and Residual Sugar

Cell dry weight determination: Filter 25 mL of the sample through a vacuum filtration device to obtain the mycelium on a pre-dried and weighed filter paper. Then, wash the filter paper five times with distilled water and dry it at 80 °C to a constant weight.

Residual sugar determination: To determine the total sugar content in the supernatants, an equal volume of 6 mol/L HCl was added, and the mixture was heated in a boiling-water bath for 30 min. The pH was then adjusted to neutral, and the sugar content was determined using the DNS method [19,20]. The DNS method was performed as described in previous reports [19,20]. Briefly, 1 mL of the diluted supernatant was mixed with 1 mL of DNS reagent and boiled for 5 min in a water bath. The mixture was then cooled to room temperature, and the absorbance at 540 nm was measured to determine the content of reducing sugars.

### 2.9. Enzyme Activity Assays of PK and PTA

Sample preparation for the enzyme activity assays of PK and PTA were performed using a modified version of the method described by Bergman et al. (2016) [21]. The strains were cultivated in a citrate fermentation medium. After being washed twice with water, the mycelia were immediately ground into a powder using liquid nitrogen. For each sample, 0.2 g of powder was collected and mixed with 1 mL of a protein extraction buffer (pH 7.0) that contained 50 mM histidine-HCl, 20 mM KH_2_PO_4_-Na_2_HPO_4_, 2 mM dithiothreitol, 1 mM MgSO_4_ and 1× Protease inhibitor mixture (Solarbio, P6730). The mixture was thoroughly blended and then centrifuged at 10,000× *g* for 10 min at 4 °C. The supernatant was stored on ice. The total protein concentrations were determined using the BCA Protein Assay Kit (TaKaRa, T9300A). The activities of PK and PTA were measured in 96-well microtiter plates using the same ferric hydroxamate assay methods described by Bergman et al. (2016) [21]. The substrate for the assays of PK activities was fructose-6-phosphate.

### 2.10. Statistical Analysis

All experiments were performed in triplicate. Mean values were compared using Student’s *t* test whenever indicated, and the difference was considered statistically significant when the *p*-value < 0.05.

## 3. Results

### 3.1. Role of ACS in the Biosynthesis of Acetyl-CoA and Citric Acid in A. niger

The addition of acetate recovered the growth defect of *acl*-deletion *A. niger* (Figure 1A), suggesting that ACS converting acetate to acetyl-CoA was likely an important supplementary pathway of cytosolic acetyl-CoA [12]. Thus, the function of *acs* (locus_tag: ASPNIDRAFT_214348) in *A. niger* S469 was explored. The expression of *acs* was detected in *A. niger* S469 during the culture process, the results of the 3 d and 5 d RT-PCR analysis are shown in Appendix A.

First, the *acs* was knocked out in S469, and the Δ*acs* mutant S688 was obtained. No obvious growth defect was observed with S688 cultured on the PDA plate (Appendix A). However, the extracellular citric acid level was reduced to 7.89 g/L in 3 d shake flask fermentation, which was a 25.7% decrease compared with the parent strain (Figure 1B). The intracellular levels of acetyl-CoA and the extracellular levels of acetic acid in S469 and S688 were also investigated in 3 d shake flask fermentation. A 11.6% decrease in acetyl-CoA (37.3 nmol/mg total protein) and an 8.8% increase in acetic acid (1.67 g/L) were observed in the strain S688 compared with the parent strain (Figure 1B). So, it can be concluded that the deletion of *acs* resulted in a decrease in the accumulation of acetyl-CoA and citric acid, with a slight increase in acetic acid in *A. niger*.

Further, the gene complementation of *acs* was performed by integrating the expression cassette of *acs* driven by the *gpdA* promoter at the *amyA* locus (ASPNIDRAFT_47911) of the genome of the Δ*acs* mutant S688, and the strain S1422 was obtained. The extracellular citric acid level and intracellular acetyl-CoA level of S1422 in 3 d shake flask fermentation reached 12.72 g/L and 47.8 nmol/mg total protein, respectively, which are obvious increases compared with S688, even higher than that of strain S469. Meanwhile, a 14.4% decrease in the extracellular acetic acid level was observed with S1422 in 3 d shake flask fermentation compared with S688.

To validate the impact of the *acs* gene on other parameters during citric acid fermentation, the cell dry weight, residual sugar and morphology of mycelial pellets were also measured. Strains S469, S688 and S1422 did not exhibit any statistically significant differences in terms of cell growth, as the cell dry weight of the three strains increased to 7.55–8.49 g/L on the third day of fermentation (Appendix A). The three strains also displayed similar levels of sugar consumption, with residual sugar ranging from 73.5 to 77.9 g/L after three days of fermentation, and no statistically significant differences were observed (Appendix A). The morphology and sizes of the mycelial pellets of the three strains were quite comparable, as demonstrated by the third-day morphology of mycelial pellets viewed under a microscope magnified by 40 times (Appendix A).

The above results indicate that the knockout and complementation of the *acs* gene had no significant effect on the cell growth and substrate consumption. However, the knockout and complementation of the *acs* had a significant impact on the citric acid yield. Theoretically, one molecule of sucrose can be converted into a maximum of two molecules of citric acid, which means the carbon atom yield of citric acid can reach 100%. The carbon atom yields of citric acid in strains S469, S688 and S1422 were 35.6%, 28.3% and 51.4%, respectively. Based on the experimental results of the gene deletion and complement of *acs*, it was verified that ACS was an important alternative pathway for cytosolic acetyl-CoA synthesis and played a crucial role in citric acid production in *A. niger*.

### 3.2. Roles of PKs and ACK in the Biosynthesis of Acetyl-CoA and Citric Acid in A. niger

Since acetic acid was very important for the biosynthesis of cytosolic acetyl-CoA in *A. niger*, its generating pathways also needed further exploration. In fact, acetic acid can be biosynthesized through many pathways. However, previous research reported that acetic acid in *A. nidulans*, which belongs to the same genus as *A. niger*, can be synthesized by PK and the acetate kinase (ACK) pathway in the cytoplasm and further converted into about 30% of the total acetyl-CoA [15]. Thus, PK and ACK may also play important roles in the cytosolic biosynthesis of acetic acid and acetyl-CoA in *A. niger* (Figure 1A). Two PKs were found in the genome of *A. niger* ATCC1015 through NCBI BLASTP, using the amino acid sequence of *A. nidulans* PK (*An*-PK) as the query sequence, namely, PKA (locus_tag: ASPNIDRAFT_197387) and PKB (locus_tag: ASPNIDRAFT_54814). The neighbor-joining phylogenetic tree of the PKA and PKB of *A. niger* ATCC1015 and 12 previously tested PKs in fungi [21,22] was constructed using MEGA software (Version 10.1.7). It showed that PKA and *An*-PK were grouped into an independent branch, while PKB was grouped into the branch of bacterial PKs (Appendix A). The locus tag of ACK in *A. niger* ATCC1015 was ASPNIDRAFT_206885. The expression of *pkA*, *pkB* and *ack* was detected in *A. niger* S469 during the culture process; the results of 3 d and 5 d RT-PCR analysis are shown in Appendix A.

The genes *ack*, *pkA* and *pkB* were knocked out in the parent strain S469, and the Δ*ack* mutant S917, Δ*pkA* mutant S2783 and Δ*pkB* mutant S2908 were obtained, respectively. The Δ*pkA*Δ*pkB* mutant S3050 was obtained by further deleting *pkA* in the Δ*pkB* mutant S2908. No obvious growth defect was observed with S917, S2783 and S2908 cultured on the PDA plate, which was consistent with the Δ*acs* mutant S688 (Appendix A). However, obvious decreases in the levels of the extracellular citric and acetic acid and intracellular acetyl-CoA of S917, S2783, S2908 and S3050 were observed compared with the parent strain S469; the results of 3 d shaker flask fermentation are shown in Figure 2A. In 3-day shake flask fermentation, S917 exhibited reductions of 20.3% and 25.1% in the extracellular levels of citric acid and acetic acid, respectively, reaching 8.47 g/L and 1.15 g/L compared with S469. Meanwhile, the intracellular level of acetyl-CoA was reduced by 21.2%, reaching 33.18 nmol/mg total protein, in S917 compared with S469. (Figure 2A). The single-knockout strains S2783 and S2908 of *pkA* and *pkB* and the double-knockout strain S3050 also showed a significant reduction in the levels of extracellular citric and acetic acid and intracellular acetyl-CoA. The decline was most pronounced in S3050, with decreases of 46.3%, 35.6% and 36.8% compared with S469, respectively (Figure 2A). Based on the above results, acetyl phosphate could be generated from the reaction catalyzed by PK and then covert into acetic acid by ACK (Figure 1A), so the deletion of *acs*, *pkA* or *pkB* in *A. niger* could result in decreases in the extracellular citric and acetic acid and intracellular acetyl-CoA, as expected.

Furthermore, the gene complementation of *ack*, *pkA* or *pkB* was performed by integrating their expression cassettes at the *amyA* locus of the respective knockout mutant genome. The transcription of the three genes in the expression cassettes was driven by the strong constitutive *gpdA* promoter. More specifically, the complementation of *ack* in S917 resulted in S1724, while the complementation of *pkA* or *pkB* in S3050 resulted in S3098 or S3104, respectively. After three days of shaking flask fermentation, the final concentrations of citric acid produced by S1724, S3098, and S3104 were 12.2 g/L, 12.6 g/L, and 15.0 g/L, respectively, higher than the citric acid concentration produced by S469 (10.6 g/L) (Figure 2B). Moreover, compared to S469, the three strains S1724, S3098 and S3104 showed an increase in the levels of intracellular acetyl-CoA by 15.5%, 10.8% and 14.7%, respectively, and accumulated more extracellular acetic acid in 3 d shake flask fermentation (Figure 2B). Based on the data above, the complementation of *ack*, *pkA* and *pkB* all increased the levels of intracellular acetyl-CoA and the biosynthesis of citric acid in *A. niger*, and the complementation of *pkB* was more efficient than that of *pkA*. The intracellular PK activity of the *pkB*-complementation strain S3104 reached 0.21 U/mg total protein, which was 1.75 times that of the *pkA*-complementation strain S3098, consistent with the conclusion in the previous sentence.

Similar to *acs*, the deletion and complementation of *ack*, *pkA* and *pkB* had no significant effect on the cell growth, sugar consumption and morphology of mycelial pellets. The cell dry weight, residual sugar and morphology on the third day of shake fermentation have been provided in the Appendix A. The deletion of *ack*, *pkA* and *pkB* led to a decrease in the carbon atom yield of citric acid to between 27.8% and 30.7%, while the complementation increases the yield to between 50.2% and 53.9%. In conclusion, all the above results indicate that PK, ACK and ACS formed an important metabolic pathway, which made a significant contribution to the biosynthesis of cytosolic acetyl-CoA (Figure 1A). By increasing the supply of cytosolic acetyl-CoA through this pathway, the conversion rate of sucrose to citric acid can be promoted, thus enhancing the production of citric acid in *A. niger*.

### 3.3. Heterologous PK-PTA Pathway Can Significantly Increase the Citrate Production of A. niger

Based on the above findings, strengthening the PK-ACK-ACS pathway by introducing more efficient PK and phosphotransacetylase (PTA) from other microorganisms may more fully provide cytosolic acetyl-CoA and thus significantly enhance the biosynthesis of citric acid.

Some PKs from different microorganisms had been evaluated in the fungi *Saccharomyces cerevisiae* and *Yarrowia lipolytica* [21,22]. In vivo, the PK enzyme activity of the *An-pk*-expression mutant was very weak in *S. cerevisiae* or not detected in *Y. lipolytica*, respectively [21,22]. In *S. cerevisiae*, the highest PK activities were found in PKs from *Bifidobacterium* strains, while in *Y. lipolytica*, it was *Ca-*PK from *Clostridium acetobutylicum* [21,22]. Therefore, *Bl*-PK (GenBank: AAN24771.1) from *B. longum*, *Ba*-PK (NCBI-Gene ID: 56674845) from *B. adolescens* and *Ca*-PK (GenBank: KHD36088.1) from *C. acetobutyricum* were selected as potential candidates for building a heterologous PK-PTA pathway in *A. niger* S469. After codon optimization, the nucleotide sequences of these three genes were synthesized and then used to construct their respective gene expression cassettes, with the transcription of the genes driven by the *gpdA* promoter. The gene expression cassettes of *Bl-pk*, *Ba- pk* and *Ca-pk* were integrated at the *amyA* locus in the genome of S3050, resulting in strains S3081, S3087 and S3092, respectively. The final citric acid concentrations of S3081, S3087 and S3092 reached 18.6, 20.1 and 21.7 g/L, respectively, in 3-day shake flask fermentation (Figure 3A). These concentrations are clearly higher than those of the *pkA*-complementation strain S3098 (12.6 g/L) and the *pkB*-complementation strain S3104 (15.0 g/L) (Figure 3A). The PK activity with fructose-6-phosphate as a substrate was also determined in S469, S3050, S3098, S3104, S3081, S3087 and S3092. The PK activities reached 0.61, 0.83 and 1.52 U/mg total protein in S3081, S3087 and S3092, respectively, which are clearly higher than those in S3098 and S3104 (Figure 3A). In the seven strains, the final titers of citric acid were correlated with the PK activities; *Ca*-PK exhibited the highest PK activity, and its expression resulted in the highest citric acid production.

Another crucial process was the conversion of acetyl-phosphate into acetyl-CoA. In bacteria, this conversion can be carried out in one step by the enzyme PTA, with less ATP consumption compared to the two-step process, catalyzed by the enzymes ACK and ACS, in *A. niger* (Figure 3B). The PTAs from *Bacillus subtilis* and *Thermoanaerobacterium saccharolyticum* showed great activity in the fungus *Y. lipolytica*, according to Kameen et al. (2021) [22]. Therefore, we selected *Ts*-PTA (GenBank: ADU15860.1) and *Bs*-PTA (NCBI-Gene ID: 936581) as potential candidates for building the heterologous PK-PTA pathway in *A. niger* S469. The strains S2909 and S2914 were created by integrating the expression cassettes of *Ts-pta* and *Bs-pta*, respectively, at the *amyA* locus of S917. When compared to strain S469, strains S2909 and S2914 both generated more citric acid, with final titers of 17.4 and 15.6 g/L, respectively (Figure 3C). They also had greater PTA activities, with respective values of 1.32 and 0.54 U/mg total protein (Figure 3C). *Ts*-PTA performed best in terms of PTA activity and citric acid generation.

According to the above results, the pLH1805 vector was created, and it contained both the *Ca-pk* and *Ts-pta* expression cassettes. Then, the strains of co-expression *Ca-pk* and *Ts-pta* were created by randomly inserting the two expression cassettes into S469 using a method of non-homologous recombination. Eleven correctly transformed strains were obtained, and the strain with the highest citric acid production was designated as S3158. Shake flask fermentation was subsequently carried out to compare the citric acid production of S3158 with that of the parental strain, S469. The results of the shake flask fermentation on the third day are presented in Figure 3D. The titer of citric acid produced by S3158 reached 24.45 g/L, representing a significant increase of 121.3% compared to that of S469. The biomasses of the S469 and S3158 strains were 8.17 g/L and 9.11 g/L, respectively, while their residual sugar levels were 68.8 g/L and 60.6 g/L, respectively. S3158 exhibited slightly stronger sugar consumption and growth abilities than S469, but the most significant difference was in its carbon atom yield of citric acid, which was 55.3%, 1.75 times higher than that of S469. In conclusion, the reconstruction of a highly efficient PK-PTA pathway in *A. niger* S469 significantly increased citric acid production by directing carbon flux from fructose-6-phosphate, an intermediate in the glycolysis pathway, to the cytosolic acetyl-CoA.

### 3.4. Citric Acid Fermentation in a 2 L Bioreactor

To investigate the performance of S5138 at the bioreactor scale, we conducted a 5-day citric acid fermentation experiment in a 2 L fermenter. The citric acid titer, biomass, residual sugar and pH were measured, and the morphology of the mycelial pellets during the fermentation process was observed (Figure 4). The most significant difference between the starting strain S469 and S3158 was the synthesis of citric acid. The titer of citric acid produced by both strains increased throughout the entire fermentation process. On the 5th day, S3158 produced 53.18 g/L of citric acid, which was 96.4% higher than that of S469. The consumption of sugar by both strains was similar, with the 5th-day residual sugar values of 44.2 g/L and 39.9 g/L for S469 and S3158, respectively. The overexpression of *pk*-*pta* did not significantly increase the metabolic flux of sugar, but it enhanced the conversion rate of sugar to citric acid. The carbon atom yield of citric acid in S3158 was 73.8%, which was 1.88 times higher than that of S469. The cell growth and pH changes during the fermentation process were similar between the two strains. Both S469 and S3158 showed rapid initial biomass growth, peaking at 48 h with values of 9.34 and 10.40 g/L, respectively. The initial pH of the fermentation medium was adjusted to around 2.5 to facilitate citric acid synthesis, and the pH rapidly decreased to below 1 during the fermentation process. The morphological changes of the mycelial pallet were observed under a microscope at a magnification of 40 times (Figure 4B). The mycelial pellets at 0 h were the seed state after 24 h of cultivation in the shaking flask. The mycelial pellets of the two strains showed no apparent differences in morphology at the same time. The diameter of the mycelial pellets of both strains rapidly increased before 48 h and remained almost unchanged after 48 h, which was consistent with the trend of biomass changes. In summary, the expression of *pk*-*pta* had little effect on the morphology of mycelial pellets, biomass growth, and sugar metabolism during the fermentation process. Its contribution to increased citric acid production was due to an enhanced conversion rate of sugar to citric acid.

## 4. Discussion

Over the past few decades, super strains of *A. niger* that produce citric acid have been developed through mutagenic breeding and are now extensively used in the commercial production of citric acid [1,2]. However, in contrast to the development and industrial application of high-yielding strains, the metabolic regulation mechanism of citric acid synthesis in *A. niger* is relatively underdeveloped. The mitochondrial biosynthetic pathway of citric acid in *A. niger* had been well established quite early on. However, recent genome sequencing and annotation of *A. niger* suggested that citric acid biosynthesis not only takes place in the mitochondria but may also occur in the cytosol [12,14]. In the *A. niger* genome, three citrate synthases have been identified, with two of them being localized in mitochondria and one in the cytosol, named CitB [14]. The heterologous expression of *citB* can induce the secretion of citric acid in non-citrate-producing *A. nidulans* [14]. These results suggested the possibility of a cytosolic biosynthesis pathway of citric acid in *A. niger*. However, this hypothesis required further support with critical evidence, such as the sources of cytosolic acetyl-CoA and oxaloacetate, the direct precursors required for cytosolic citrate biosynthesis. The findings from this study suggested that the PK-ACK-ACS pathway, involving *pkA*, *pkB*, *ack* and *acs*, played an important complementary role in the biosynthesis of cytosolic acetyl-CoA. Working in conjunction with the well-known ACL pathway, the PK-ACK-ACS pathway contributed to the level of cytosolic acetyl-CoA and exerted a significant influence on citric acid biosynthesis (Figure 1A). Thus, the PK-ACK-ACS pathway in *A. niger* can provide additional acetyl-CoA for the cytosolic biosynthesis of citric acid.

Knocking out any of the genes *pkA*, *pkB*, *ack* and *acs* resulted in decreased citric acid production, whereas overexpression led to an increase. However, neither the knockout nor overexpression of these genes caused significant changes in the growth, sugar metabolism or mycelial morphology of *A. niger*. Instead, they altered the yield of citric acid from sucrose, thereby affecting overall citric acid production. Based on these results, we formulated the following two speculations. First, the PK-ACK-ACS pathway maybe minimizes carbon loss, leading to an increased yield of citric acid. Normally, acetyl-CoA was generated from pyruvate via pyruvate dehydrogenase, resulting in a maximum of two acetyl-CoA molecules from one glucose molecule with one-third carbon loss. The PK-ACK-ACS pathway had been widely used in metabolic engineering to improve the yield of acetyl-CoA in bacteria, yeast and other microorganisms by reducing carbon loss [23]. Previous studies had demonstrated that carbon rearrangement cycles based on PK can convert one molecule of glucose into three molecules of acetyl-CoA [24,25,26,27]. The existence of two types of PK in *A. niger* indicated that this type of carbon rearrangement cycle may be occurring. Our calculations indicated that the maximum theoretical yield of citric acid from glucose through the glycolytic pathway alone was 1.0 moL/moL glucose, while the theoretical yield of citric acid can reach 1.2 moL/moL glucose through the carbon rearrangement cycles (Appendix A). Second, compared to the glycolysis pathway, the PK-ACK-ACS pathway produced less NADH and ATP during the conversion from fructose 6-phosphate to acetyl-CoA, which was more favorable for citric acid accumulation [4,8]. Energy metabolism played a crucial role in the biosynthesis of citric acid in *A. niger* [4,8]. The AOX enabled the bypassing of the electron transport chain in the mitochondria, allowing for direct oxidation of NADH and generating heat instead of ATP. Research had shown that the overexpression of AOX in *A. niger* can increase the production of citric acid [4,8]. Although the AOX pathway and the PK-ACK-ACS pathway had distinct reactions, they both could contribute to regulating energy metabolism in a similar manner. These are likely the cause of the significant increase in the citric acid yield achieved by constructing an efficient heterologous PK-PTA pathway.

The above results and analysis supported the viewpoint that *A. niger* had the cytosolic citric acid biosynthesis pathway, but the evidence was not yet sufficient. This was because cytosolic acetyl-CoA was mainly produced by the cleavage of citrate via ACL (Figure 1A). The overexpression of the PK-ACK-ACS pathway may provide more cytosolic acetyl-CoA, which could weaken citrate cleavage and indirectly increase citric acid production. Further investigation is still necessary to definitively determine the function of the cytosolic citrate synthesis pathway in the citric acid production of *A. niger*. Primarily, while the function of CitB had been established in *A. nidulans*, its significance in the citric acid production of *A. niger* has not been fully clarified. Therefore, in our future work, we plan to conduct complement and knockout experiments on *citB* to further elucidate its role. Additionally, cytosolic acetyl-CoA could be generated through various other pathways, aside from the PK-ACK-ACS pathway confirmed in this work. The conversion of pyruvate to acetic acid through pyruvate decarboxylase and acetaldehyde dehydrogenase is a well-known pathway in many microorganisms, including *A. niger* [3,28]. We also found that lactic acid may be converted into acetic acid through the action of lactic acid dehydrogenase (NCBI-Gene ID: 4977180 and 4986178) in the cytosol of *A. niger*. Therefore, a comprehensive investigation into the metabolic pathways involved in the generation of cytosolic acetyl-CoA would be valuable in understanding the metabolic mechanism of citric acid in *A. niger*.

So far, various studies have been conducted on the metabolic engineering of *A. niger* to enhance citric acid production, all of which are based on the understanding of mitochondrial citrate biosynthetic pathways. These studies included precursor supplement pathway engineering, feedback inhibition engineering, respiratory chain regulation, citrate transport engineering and so on [1,2,3,4,5]. The metabolic engineering of citrate transport and the respiratory chain have been found to significantly improve citric acid production [6,7,8]. Interestingly, simply overexpressing mitochondrial citrate synthase and enhancing the availability of oxaloacetate or acetyl-CoA may not be sufficient to improve citric acid biosynthesis [4,9,10,11,12]. The bioproduction of citric acid in *A. niger* appears to involve a complex regulatory process [4]. These reported metabolic engineering strategies may not have a significant effect on the improvement of high-performance industrial strains of *A. niger* currently in use. This study can help in better understanding the metabolic pathways and regulatory mechanisms of citric acid in *A. niger* and also open new possibilities for the metabolic engineering of industrial *A. niger* strains.

## 5. Conclusions

In this study, we explored the role of cytosolic acetyl-CoA and its biosynthesis pathway in citric acid production in *A. niger*. Our results revealed that the PK-ACK-ACS pathway served as a critical acetyl-CoA supplement pathway, significantly impacting the production and yield of citric acid. This pathway could provide a direct precursor for cytosolic citric acid biosynthesis with minimal carbon loss and energy generation (NADH and ATP). Based on these new insights, we introduced an efficient heterologous PK-PTA pathway to reinforce the PK-ACK-ACS pathway in *A. niger*, resulting in an increase of 96.4% in citric acid production and of 88% in the yield during bioreactor fermentation. Our results suggest that a cytosolic citrate biosynthesis pathway may exist in *A. niger* and play a crucial role in the citric acid production. These findings not only deepen our understanding of the metabolic mechanism of citric acid and acetyl-CoA in *A. niger* but also offer a novel perspective for the metabolic engineering of industrial *A. niger* strains to improve citric acid production.

## Figures and Tables

**Figure 2 jof-09-00504-f002:**
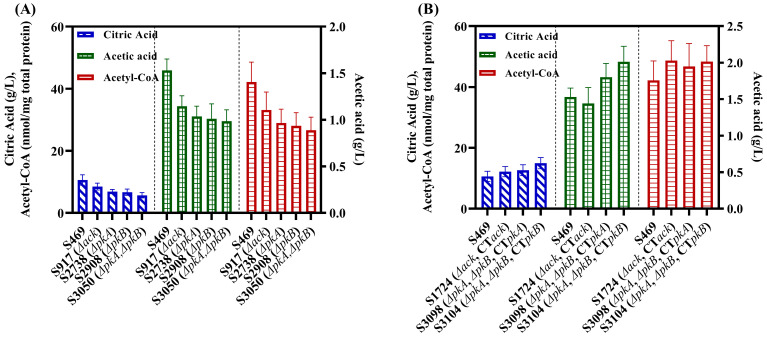
Roles of ACK, PKA and PKB in the biosynthesis of acetyl-CoA and citric acid of *A. niger*. (**A**) the concentrations of extracellular citric and acetic acid and intracellular acetyl-CoA of the S469, S917 (Δ*ack*), S2738 (Δ*pkA*), S2908 (Δ*pkB*) and S3050 (Δ*pkA*Δ*pkB*) during 3-day citrate fermentation in a shake flask at 28 °C and 200 rpm; (**B**) the concentrations of extracellular citric and acetic acid and intracellular acetyl-CoA of the S469, S1724 (Δ*ack*, *CTack*), S3098 (Δ*pkA*Δ*pkB*, *CTpkA*) and S3104 (Δ*pkA*Δ*pkB*,*CTpkB*) during 3-day citrate fermentation in a shake flask at 28 °C and 200 rpm. *CT*: complementation of genes driven by the *gpdA* promoter at the *amyA* locus.

**Figure 3 jof-09-00504-f003:**
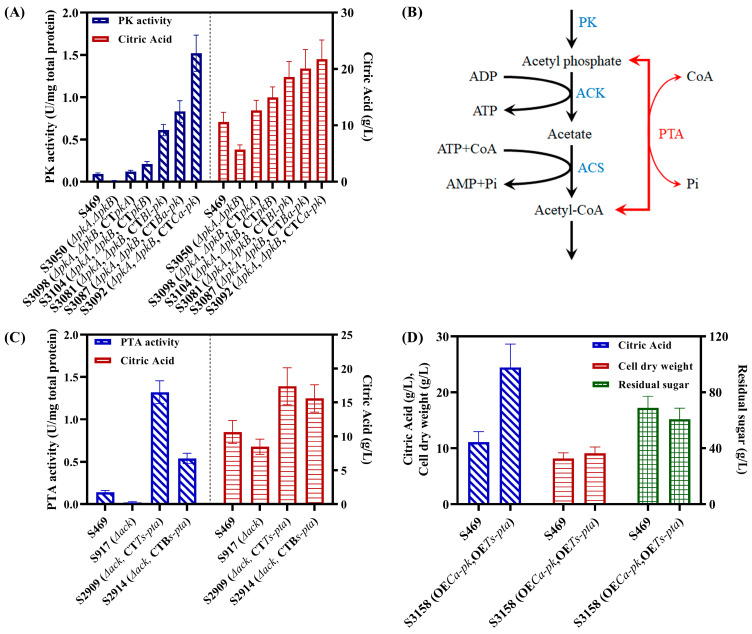
Reconstruction of the heterologous PK-PTA pathway in *A. niger* S469. (**A**) The PK activities and citric acid production of S469, S3050 (Δ*pkA*Δ*pkB*), S3098 (Δ*pkA*Δ*pkB*, *CTpkA*), S3104 (Δ*pkA*Δ*pkB*,*CTpkB*), S3081 (Δ*pkA*Δ*pkB*, *CTBl-pk*), S3087 (Δ*pkA*Δ*pkB*, *CTBa-pk*) and S3092 (Δ*pkA*Δ*pkB*, *CTCa-pk*) for screening an efficient PK used for the construction of a heterologous PK-PTA pathway; (**B**) the reactions of the native ACK-ACS pathway of *A. niger* and heterologous PTA from bacteria; (**C**) the PTA activities and citric acid production of S469, S917 (Δ*ack*), S2909 (Δ*ack*, *CTTs-pta*) and S2914 (Δ*ack*, *CTBs-pta*) for screening an efficient PTA used for the construction of a heterologous PK-PTA pathway; (**D**) shake flask citrate fermentation of *A. niger* S469 and S3158 (*OECa-pk*, *OETs-pta*) at 28 °C for 3 d. Note: the activities of PK and PTA were assayed in shake flask citrate fermentation at 28 °C for 3 d. *CT*: complementation of genes driven by the *gpdA* promoter at the amyA locus. *OE*: overexpression of *Ca-pk* and *Ts-pta* in S469.

**Figure 4 jof-09-00504-f004:**
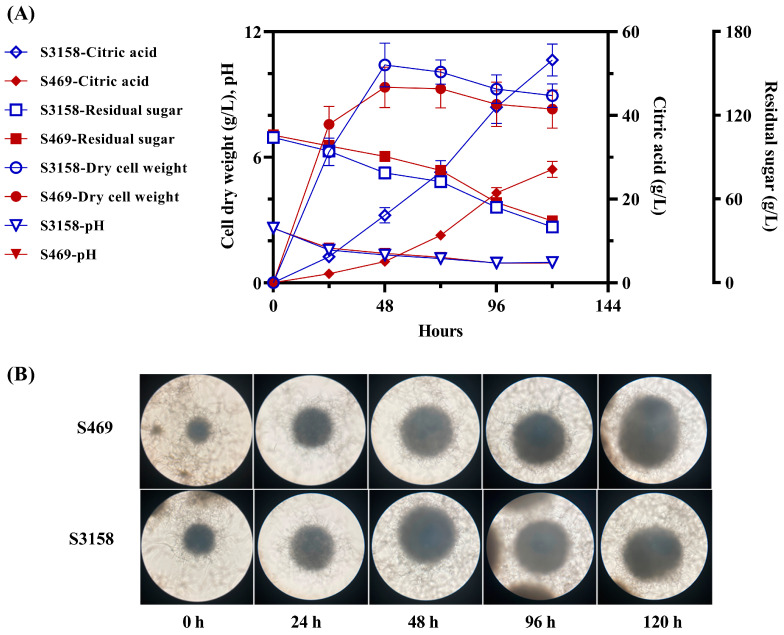
Citric acid fermentation of *A. niger* S469 and S3158 (OE*Ca-pk*, OE*Ts-pta*) in a 2 L bioreactor. (**A**) The citric acid yield, cell dry weight, residual sugar and pH in the citrate fermentation of the 2 L bioreactor at 28 °C for 120 h. (**B**) Mycelial morphology of S469 and S3158 in the citrate fermentation of the 2 L bioreactor at 28 °C for 120 h.

**Table 1 jof-09-00504-t001:** *A. niger* strains and plasmids used in this study.

	Genotype/Description	Source
Strains		
S469	Tet-On::*cre*, *hyg^s^*	Our laboratory [16]
S687	Tet-On::*cre*, Δ*acs*, *hyg^r^*	This study
S688	Tet-On::*cre*, Δ*acs*, *hyg^s^*	This study
S916	Tet-On::*cre*, Δ*ack*, *hyg^r^*	This study
S917	Tet-On::*cre*, Δ*ack*, *hyg^s^*	This study
S2738	Tet-On::*cre*, Δ*pkA*, *hyg^r^*	This study
S2740	Tet-On::*cre*, Δ*pkB*, *hyg^r^*	This study
S2908	Tet-On::*cre*, Δ*pkB*, *hyg^s^*	This study
S3008	Tet-On::*cre*, Δ*pkB,* Δ*pkA*, *hyg^r^*	This study
S3050	Tet-On::*cre*, Δ*pkB,* Δ*pkA*, *hyg^s^*	This study
S1422	Tet-On::*cre,* Δ*acs*, *amyA*::*PgpdA*::*acs*::*TtrpC*, *hyg^r^*	This study
S1724	Tet-On::*cre*, Δ*ack*, *amyA*::*PgpdA*::*ack*::*TtrpC*, *hyg^r^*	This study
S3081	Tet-On::*cre*, Δ*pkB*, Δ*pkA*, *amyA*::*PgpdA*::*Bl-pk*::*TtrpC*, *hyg^r^*	This study
S3087	Tet-On::*cre*, Δ*pkB*, Δ*pkA*, *amyA*::*PgpdA*::*Ba-pk*::*TtrpC*, *hyg^r^*	This study
S3092	Tet-On::*cre*, Δ*pkB*, Δ*pkA*, *amyA*::*PgpdA*::*Ca-pk*::*TtrpC*, *hyg^r^*	This study
S3098	Tet-On::*cre*, Δ*pkB*, Δ*pkA*, *amyA*::*PgpdA*::*pkA*::*TtrpC*, *hyg^r^*	This study
S3104	Tet-On::*cre*, Δ*pkB*, Δ*pkA*, *amyA*::*PgpdA*::*pkB*::*TtrpC*, *hyg^r^*	This study
S2909	Tet-On::*cre*, Δ*ack*, *amyA*::*PgpdA*::*Ts-pta*::*TtrpC*, *hyg^r^*	This study
S2914	Tet-On::*cre*, Δ*ack*, *amyA*::*PgpdA*::*Bs-pta*::*TtrpC*, *hyg^r^*	This study
S3158	Tet-On::*cre*, *PgpdA*::*Ca-pk*::*TtrpC*, *PgpdA*::*Ts-pta*::*TtrpC*, *hyg^r^*	This study
Plasmids		
pLH454	*lox*P*-hph-lox*P, *hyg^r^, kan^r^, PgpdA*, *TtrpC*	Our laboratory [16]
pLH594	*lox*P*-hph-lox*P*, hyg^r^, ppt^r^, kan^r^*	Our laboratory [16]
pLH924	*lox*P*-hph-lox*P*, hyg^r^, ppt^r^, kan^r^,* Δ*amyA*	Our laboratory [17]
pLH525	*lox*P*-hph-lox*P*, hyg^r^, ppt^r^, kan^r^,* Δ*acs*	This study
pLH670	*lox*P*-hph-lox*P*, hyg^r^, ppt^r^, kan^r^,* Δ*ack*	This study
pLH1549	*lox*P*-hph-lox*P*, hyg^r^, ppt^r^, kan^r^,* Δ*pkA*	This study
pLH1551	*lox*P*-hph-lox*P*, hyg^r^, ppt^r^, kan^r^,* Δ*pkB*	This study
pLH821	*lox*P*-hph-lox*P, *hyg^r^, kan^r^, PgpdA*::*acs*::*TtrpC*	This study
pLH834	*lox*P*-hph-lox*P, *hyg^r^, kan^r^, PgpdA*::*ack*::*TtrpC*	This study
pLH1587	*lox*P*-hph-lox*P, *hyg^r^, kan^r^, PgpdA*::*Bl-pk*::*TtrpC*	This study
pLH1588	*lox*P*-hph-lox*P, *hyg^r^, kan^r^, PgpdA*::*Ba-pk*::*TtrpC*	This study
pLH1610	*lox*P*-hph-lox*P, *hyg^r^, kan^r^, PgpdA*::*Ca-pk*::*TtrpC*	This study
pLH1628	*lox*P*-hph-lox*P, *hyg^r^, kan^r^, PgpdA*::*pkA*::*TtrpC*	This study
pLH1630	*lox*P*-hph-lox*P, *hyg^r^, kan^r^, PgpdA*::*pkB*::*TtrpC*	This study
pLH1589	*lox*P*-hph-lox*P, *hyg^r^, kan^r^, PgpdA*::*Ts-pta*::*TtrpC*	This study
pLH1590	*lox*P*-hph-lox*P, *hyg^r^, kan^r^, PgpdA*::*Bs-pta*::*TtrpC*	This study
pLH822	*lox*P*-hph-lox*P*, hyg^r^, ppt^r^, kan^r^,* Δ*amyA, PgpdA*::*acs*::*TtrpC*	This study
pLH835	*lox*P*-hph-lox*P*, hyg^r^, ppt^r^, kan^r^,* Δ*amyA, PgpdA*::*ack*::*TtrpC*	This study
pLH1591	*lox*P*-hph-lox*P*, hyg^r^, ppt^r^, kan^r^,* Δ*amyA, PgpdA*::*Bl-pk*::*TtrpC*	This study
pLH1592	*lox*P*-hph-lox*P*, hyg^r^, ppt^r^, kan^r^,* Δ*amyA, PgpdA*::*Ba-pk*::*TtrpC*	This study
pLH1611	*lox*P*-hph-lox*P*, hyg^r^, ppt^r^, kan^r^,* Δ*amyA, PgpdA*::*Ca-pk*::*TtrpC*	This study
pLH1629	*lox*P*-hph-lox*P*, hyg^r^, ppt^r^, kan^r^,* Δ*amyA, PgpdA*::*pkA*::*TtrpC*	This study
pLH1631	*lox*P*-hph-lox*P*, hyg^r^, ppt^r^, kan^r^,* Δ*amyA, PgpdA*::*pkB*::*TtrpC*	This study
pLH1593	*lox*P*-hph-lox*P*, hyg^r^, ppt^r^, kan^r^,* Δ*amyA, PgpdA*::*Ts-pta*::*TtrpC*	This study
pLH1594	*lox*P*-hph-lox*P*, hyg^r^, ppt^r^, kan^r^,* Δ*amyA, PgpdA*::*Bs-pta*::*TtrpC*	This study
pLH1080	*lox*P*-hph-lox*P, *hyg^r^, kan^r^*, *PgdhA*, *TtrpC*	This study
pLH1804	*lox*P*-hph-lox*P, *hyg^r^, kan^r^, PgdhA*::*Ca-pk*::*TtrpC*	This study
pLH1805	*lox*P*-hph-lox*P*, hyg^r^, ppt^r^, kan^r^, PgdhA::Ca-pk*::*TtrpC, PgpdA::Ts-pta*::*TtrpC*	This study

*hyg^r^*: hygromycin B resistance; *hyg^s^*: hygromycin B-sensitive; *ppt^r^*: phosphinothricin resistance; *kan^r^*: kanamycin resistance.

## Data Availability

The data presented in this study are available upon request from the corresponding author.

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
