# Peer review of "Engineering a Phosphoketolase Pathway to Supplement Cytosolic Acetyl-CoA in Aspergillus niger Enables a Significant Increase in Citric Acid Production"

_jof, 2023, doi:10.3390/jof9050504_

Round 1

Reviewer 1 Report

The manuscript on Engineering Phosphoketolase Pathway to Supplement Cyto- 2 solic Acetyl-CoA in Aspergillus niger Enables Significant In- 3 crease of Citric Acid Production is written well and presented well. 

In conclusion, the availability of cytosolic acetyl-CoA is very important for citric acid 441 biosynthesis in A. niger. PK, ACK, and ACS act as a complementary pathways in the bio- 442 synthesis of cytosolic acetyl-CoA and contributed greatly to the citric acid biosynthesis in 443 A. nizer. 

The suggestion needs to be incorporated 

1. Introduction and Discussion should be improved by adding more references to this section. The discussion is not written properly. 

Author Response

Response to Reviewer 1 Comments

Comments and Suggestions for Authors: Introduction and Discussion should be improved by adding more references to this section. The discussion is not written properly.

Response: Thank you very much for the comments. We have carefully reviewed the literature on citric acid production by Aspergillus niger and added a total of 7 references in the revised Introduction, Materials and Methods, and Discussion sections. These references include studies and reviews on metabolic engineering of A. niger, and the newly added residual sugar measurement method. We have partially revised the Introduction and Materials and Methods sections based on the literature. In response to the reviewer's comments, we have reorganized the content and logic of the Discussion section and rewritten it. The revised Discussion section can be found in lines 477~558 of the revised manuscript. All the changes are highlighted in yellow background.

Reviewer 2 Report

Reviewer report for paper entitled "Engineering Phosphoketolase Pathway to Supplement Cytosolic Acetyl-CoA in Aspergillus niger Enables Significant Increase of Citric Acid Production". This paper was focused on the genetically modified strains of the recombinant A. niger S469 for further reengineering of Phosphoketolase pathway. The idea of this research of interest. However, many drawbacks need to be considered before publication of this work as follows:

1. The results presented was focused on data of simple metabolites activities after just preliminary screening in small scale level for 3-5 days. Data only for citric acid, acetic acid, and acetyl-CoA. No data about the cell dry weight, pH and other important kinetic parameters, how it shifted from the original strain. Complete growth curve kinetics for all strains need to be provided (including pH, substrate consumption, metabolites production, and growth morphology). Without these data discussion will be difficult to prove the advantages of the newly constructed strain. 

2- No data was provided about the fungal morphology in submerged culture. In most cases of A. niger cultivation in submerged culture, if any genetic or cultural condition changes carried out, growth morphology need to be provided. In most cases, the growth morphological changes is the bottle neck which govern metabolites production.

3- For this type of strain modification to confirm strain overproduction, cultivation need to be run bioreactor level as well to compare all growth kinetic data between the parent strain and the genetically modified strains under both pH controlled and pH uncontolled conditions.

Without these basic data, this work is considered as preliminary research without significant added value for researchers work in this field. 

Author Response

Response to Reviewer 2 Comments

Thanks for reviewer’s valuable feedback. We have analyzed other parameters (cell dry weight, residual sugar, and mycelial particle phenotype) of all strains in the fermentations, which will help us better understand the characteristics of the strains. Data analysis indicates that neither knockout nor overexpression of these genes cause significant changes in the growth, sugar metabolism, or mycelial morphology of A. niger. Instead, they altered the yield of citric acid from sucrose, thereby affecting overall citric acid production. The findings have already been presented in the revised manuscript, and we hope that these results and discoveries will address the concerns of the reviewer. More detailed response is as follows.

Comment 1: The results presented was focused on data of simple metabolites activities after just preliminary screening in small scale level for 3-5 days. Data only for citric acid, acetic acid, and acetyl-CoA. No data about the cell dry weight, pH and other important kinetic parameters, how it shifted from the original strain. Complete growth curve kinetics for all strains need to be provided (including pH, substrate consumption, metabolites production, and growth morphology). Without these data discussion will be difficult to prove the advantages of the newly constructed strain.

Response 1: Based on the reviewer's comment, we have re-examined the fermentation data of all strains and found that their kinetics curves are quite similar. Therefore, we have added data on cell dry weight and residual sugar on the third day of fermentation for all strains to characterize their fermentation characteristics. The added data can be found in Figure S5 and Figure S6 of the revised supplementary material. We also observed the morphology of the strains with acs, ack, pkA, and pkB knocked out or complemented, but no significant differences were found. The mycelial pellets’ morphology of these strains on the third day of shake flask fermentation is shown in Figure S4 of the revised supplementary material. The results section has been updated to analyze the added data, which can be found in lines 279~298 and lines 353~363 of the revised manuscript. pH value has a significant impact on citric acid fermentation, and A. niger can efficiently synthesize citric acid only under low pH conditions. Therefore, the initial pH value of our citric acid fermentation medium was set at 2.5. The pH value of all fermentation broths of the strains rapidly decreased to around 0.95 within 48 hours and remained stable, and there was no significant difference among them. Therefore, we do not include the pH data of the shake flask fermentations in the figures, but we analyzed the pH changes during the bioreactor fermentation, which can be found in lines 454~459 of the revised manuscript.

Comment 2: No data was provided about the fungal morphology in submerged culture. In most cases of A. niger cultivation in submerged culture, if any genetic or cultural condition changes carried out, growth morphology need to be provided. In most cases, the growth morphological changes is the bottle neck which govern metabolites production.

Response 2: Thank you for the valuable feedback. As you rightly pointed out, fungal morphology plays an important role in the production of citric acid in submerged culture. The morphology of fungal mycelia is affected by multiple factors, such as pH and composition of the culture medium, as well as gene expression. We have examined the impact of genetic modifications on the morphology of A. niger in both shake flask and bioreactor fermentations, but we did not observe significant changes in the morphology of the mycelia when compared to the original strain. We have now included data of the mycelial pellets’ morphology of the strains (acs, ack, pkA, and pkB knocked out or complemented) in Figure S4 of the revised supplementary material. The mycelial pellets’ morphology of S469 and S3158 in bioreactor fermentation was also provided in Figure 4 of the revised manuscript.

Comment 3: For this type of strain modification to confirm strain overproduction, cultivation need to be run bioreactor level as well to compare all growth kinetic data between the parent strain and the genetically modified strains under both pH controlled and pH uncontolled conditions.

Response 3: Thank you for the suggestion. We have conducted three batches of bioreactor fermentation experiments, and the experimental methods are provided in lines 195~200 of the revised manuscript. As we have not specifically optimized the bioreactor fermentation of Aspergillus niger ATCC1015 for citric acid production, the medium used in the bioreactor fermentation was same as that used in shake flask fermentation, and other operation parameters were set based on the experience provided by SHANDONG ENSIGN INDUSTRY CO., LTD. (a large enterprise specializing in the production of citric acid). The fermentation of A. niger for citric acid production needs to be conducted under low pH conditions, so the initial pH value of the fermentation medium was adjusted to 2.5, and the pH value was stabilized at around 0.95 after 48 hours. The kinetic data of the bioreactor fermentation experiments are shown in Figure 4 of the revised manuscript, and the results analysis can be found in lines 442~468 of the revised manuscript.

Reviewer 3 Report

Liu and colleagues describe in their paper "Engineering Phosphoketolase Pathway to Supplement Cyto-2 solic Acetyl-CoA in Aspergillus niger Enables Significant Increase of Citric Acid Production" a very sound piece of research. They were able to significantly increase the production of citrate by more than 110% and thus provide compelling evidence for the importance of the cytosolic citrate biosynthesis pathway. 

The research appears soon to me. I have no major concerns. However some of the figures (1C, 3B, 3C, 4A, 4C) look very old fashioned. I would change that. In addition I think, that Fig. 2A is not really required. 

Author Response

Response to Reviewer 3 Comments

Comments and Suggestions for Authors: Liu and colleagues describe in their paper "Engineering Phosphoketolase Pathway to Supplement Cyto-2 solic Acetyl-CoA in Aspergillus niger Enables Significant Increase of Citric Acid Production" a very sound piece of research. They were able to significantly increase the production of citrate by more than 110% and thus provide compelling evidence for the importance of the cytosolic citrate biosynthesis pathway.

The research appears soon to me. I have no major concerns. However some of the figures (1C, 3B, 3C, 4A, 4C) look very old fashioned. I would change that. In addition I think, that Fig. 2A is not really required.

Response: We appreciate your positive feedback on our research. We hope the changes we have made will address your concerns. They are as follows:

(1) We have moved the original Figure 2 to the supplementary materials, and it is now labeled as Figure S2.

(2) The colony images from the original Figures 1 and 2 have been integrated into Figure S3.

(3) We have also added a new Figure S4, S5 and S6 in the supplemental materials to describe the dry cell weight, residual sugar, and morphology of mycelial pellets of relevant strains on the third day of shake flask fermentation.

(4) We have updated the figures 1, 2, 3 and 4 in the revised manuscript to give them a more modern appearance, as suggested.

Round 2

Reviewer 2 Report

Authors did significant improvement in manuscript quality. They included important and significant data and discussed the work properly. However, authors need to give conclusion section to address the novelty and potential application of the results obtained. Therefore, I recommend accepting this work after minor revision. 

Author Response

Response to Reviewer 2 Comments

Comment: Authors did significant improvement in manuscript quality. They included important and significant data and discussed the work properly. However, authors need to give conclusion section to address the novelty and potential application of the results obtained. Therefore, I recommend accepting this work after minor revision. 

Response:

Thank you for your constructive comments on our manuscript. We are pleased to hear that you found our work to be significantly improved with the inclusion of important data and proper discussion.

Based on your suggestion, we have included a conclusion section in our manuscript to address the novelty and potential applications of our results. In this section, we have summarized the main findings of our study and discussed their significance and practical implications. The conclusion section has been updated in lines 559~572 of the revised manuscript.

We appreciate your insightful comments and suggestions, which have helped us to improve the quality of our manuscript. We hope that our revised manuscript meets your expectations and look forward to hearing from you soon.
